# IGNN-SOLVER: A GRAPH NEURAL SOLVER FOR IMPLICIT GRAPH NEURAL NETWORKS

## ABSTRACT

Implicit graph neural networks (IGNNs), which exhibit strong expressive power with a single layer, have recently demonstrated remarkable performance in capturing long-range dependencies (LRD) in underlying graphs while effectively mitigating the over-smoothing problem. However, IGNNs rely on computationally expensive fixed-point iterations, which lead to significant speed and scalability limitations, hindering their application to large-scale graphs. To achieve fast fixed-point solving for IGNNs, we propose a novel graph neural solver, IGNN-Solver, which leverages the generalized Anderson Acceleration method, parameterized by a small GNN, and learns iterative updates as a graph-dependent temporal process. Extensive experiments demonstrate that the IGNN-Solver significantly accelerates inference, achieving a $1.5\times$ to $8\times$ speedup without sacrificing accuracy. Moreover, this advantage becomes increasingly pronounced as the graph scale grows, facilitating its large-scale deployment in real-world applications.

## 1 INTRODUCTION

Implicit graph neural networks (IGNNs) [20; 33; 7] have emerged as a significant advancement in graph learning frameworks. Unlike traditional graph neural networks (GNNs) that stack multiple explicit layers, IGNNs utilize a single implicit layer formulated as a fixed-point equation. The solution to this fixed-point equation, known as the equilibrium, is equivalent to the output obtained by iterating an explicit layer infinitely. This allows an implicit layer to access infinite hops of neighbors, providing IGNNs with global receptive fields within just one layer [12]. As a result, IGNNs effectively address the long-standing issue of over-smoothing in conventional explicit GNNs and capture of long-range dependencies in graph-structured data [30; 1; 56].

However, existing IGNNs suffer from slow speeds and have difficulty scaling to large-scale graphs [34; 18; 52; 39]. This is primarily because IGNNs derive features by solving for fixed points, demanding substantial computational resources. For example, even on the Citeseer dataset [26] classification task with a small-scale graph, IGNNs require more than 20 forward iterative computations to nearly converge to a fixed point [20]. The computational overhead of solving fixed points is amplified by the task scale, resulting in notably slow inference speeds compared to explicit GNNs. This substantial drawback poses challenges for IGNNs in generalizing to large-scale graphs in practical scenarios.

In response to this challenge, we propose a novel graph neural solver for IGNNs, termed IGNN-Solver. It takes the IGNN layer as input, which includes the graph information matrix and layer parameters, and outputs the solution to the corresponding fixed-point equation. Unlike conventional solvers relying on root-finding algorithms like Broyden's method [2], which compute output features through iterative forward passes in a potentially large GNN, IGNN-Solver offers a distinct advantage by predicting the next iteration step via a *tiny* graph network. The proposed IGNN-Solver remarkably accelerates the model's inference speed without compromising accuracy and with only a slight increase in training overhead. This advantage becomes increasingly pronounced as the scale of the graph grows, making it particularly beneficial for deploying IGNNs in large-scale graph tasks.

Our IGNN-Solver comprises two components. First, we introduce a learnable initializer that estimates an optimal initial point for the optimization process. Second, we propose a generalized version of Anderson Acceleration (AA) [2], employing a tiny graph network to model the iterative updates as a sequence of graph-dependent steps. Compared to the solvers proposed for conventional Implicit Neural Networks (INNs) [4; 5; 6], we introduce novel improvements: learning solver parameters

through a GNN-based method. This approach circumvents the potential loss of graph information, thereby improving the model's performance. Moreover, IGNN-Solver has significantly fewer parameters compared to IGNN, and its training is independent of the IGNN's inference. Consequently, the training of IGNN-Solver proceeds rapidly without sacrificing generalization.

In our experiments, we apply IGNN-Solver to 9 real-world datasets from diverse domains and scales, including 4 large-scale datasets: Amazon-all [51], Reddit [21], ogbn-arxiv [22] and ogbn-products [22]. Our results demonstrate that the IGNN-Solver achieves higher accuracy with reduced inference time, showing a $1.5\times$ to $8\times$ speedup, and incurs minimal additional training overhead, constituting only $1\%$ of the IGNN training time.

Our main contributions are summarized as follows:

- We introduce IGNN-Solver, a method designed to predict the next fixed-point iteration of the IGNN layer via a tiny graph network. This innovative approach mitigates the need for extensive iterations, common in conventional IGNNs, thereby substantially accelerating inference while preserving accuracy and minimizing parameter consumption.

- Compared to conventional solvers proposed for INNs, we have made a novel improvement in our solver by introducing a tiny GNN to learn the parameters. This not only prevents the potential loss of graph information, but also maintains the simplicity and lightweight characteristics of neural solvers.

- We validate our approach through extensive experiments on 9 real-world datasets, including 3 large-scale. Our results demonstrate that IGNN-Solver incurs minimal computational overhead (approximately 1% of the total) and achieves up to a 8-fold increase in inference speed without compromising accuracy.

## 2 RELATED WORK

The typical GNN [26] and its variants [42; 21] have been widely used for graph data modeling in various tasks. Different GNNs have been proposed to utilize attention mechanism [42], neighbors sampling [21], pseudo-coordinates [38] and graph fusion [45]. However, due to issues such as over-smoothing, depth, and bottlenecks, these models typically involve finite aggregation layers [30; 1; 56]. To address it, recent works [20; 33; 7] have developed Implicit Graph Neural Networks, encouraging these models to capture long-range dependencies on graphs. Here, we highlight the contributions of our proposed IGNN-Solver through a detailed comparison with Implicit Graph Neural Networks (IGNNs) and Deep Equilibrium Models (DEQs), both of which are closely related to our approach.

**Implicit Graph Neural Networks.**  Instead of stacking a series of operators hierarchically, implicit GNNs define their outputs as solutions to nonlinear dynamical systems, which is initially introduced by [20] to tackle challenges associated with learning long-range dependencies in graphs. [34] proposes a new implicit graph model enabling mini-batch training without sacrificing the ability to capture long-range dependencies. Subsequently, [43] introduces a novel approach based on implicit layer to model multi-scale structures on graphs. Additionally, [7] theoretically investigates the well-posedness of the IGNN model from a monotone operator viewpoint.

Although the aforementioned IGNN works well by alleviating the problem of over-smoothing of features by allowing meaningful fixed points to propagate implicitly, the inherent *slow inference speed* of implicit networks poses a major obstacle to its scalability. The main reason is that the solver of the fixed-point network is inefficient (e.g., the Picard solver used by IGNN [20] and Anderson acceleration solver used by MIGNN [7].), makes the overhead of the fixed-point solver magnified by the task scales. In comparison, our proposed IGNN-Solver accelerates the iteration process of IGNNs, which addresses *the most limiting drawback* compared to traditional feedforward models.

Another class of IGNNs based on Neural ODEs [13] has emerged to address issues like depth and bottlenecks. For example, [3] models continuous residual layers using GCNs. [40] proposes methods for modeling static and dynamic graphs with GCNs, along with a hybrid approach where latent states evolve continuously between RNN steps in dynamic graphs. [49] tackles the problem of continuous message passing. [10] introduces a graph neural diffusion network based on the discretization of diffusion PDEs on graphs. [41] enhances graph neural diffusion with a source term and connects the

model to random walk formulation on graphs. However, they essentially view deep learning on graphs as a continuous diffusion process, which differs from IGNNs based on fixed-point networks, thus requiring manually tuning the termination time and step size of the diffusion equation. In contrast, our IGNN-Solver uses implicit formulations instead of explicit diffusion discretization, which admits an equilibrium corresponding to infinite diffusion steps and expands the receptive field.

**Fixed-point Solvers for Deep Equilibrium Models.** Traditional deep learning models use multi-layered networks, while DEQs [4] find a fixed-point of a single layer, representing the network's equilibrium state. Viewed as infinitely deep networks, DEQs use root-finding for forward propagation and implicit differentiation for backward propagation. Therefore, DEQs capture complex patterns with constant memory and computational costs [31]. Monotone operator theory has been used to guarantee the convergence of DEQs [46] and to improve the stability of implicit networks [6].

However, it is well-known that DEQs, as typical implicit models, suffer from slow training and inference speeds [32; 34], which is highly disadvantageous for large-scale scenarios like GraphGPT [58]. To alleviate this issue, recent efforts have explored certain improved solver methods for DEQs, further optimizing root-finding problems and making these models easier to solve: [23] discusses the amortization of the cost of the iterative solver that would otherwise make implicit models slow. [18] proposes a novel gradient estimate for implicit models, named phantom gradient, which significantly accelerates the backward passes in training implicit models. [6] discusses the superiority of learnable solvers over generic solvers in implicit models using a tiny neural network and significantly enhances the efficiency of such models through custom neural solvers.

In contrast, IGNN has a similar network representation to DEQ, both defining the output as *the solution of the equation* to obtain network outputs. But the difference lies in the fact that IGNN's equilibrium equations encode graph structure while DEQ does not, which will undoubtedly deepen its weakness of *slow inference speed* and make IGNN's solution slower. Insight on it, our proposed IGNN-Solver leverages graph information to guide solver acceleration, achieving *fast and meaningful* implicit graph network propagation, especially in the case of graph data being large-scale.

## 3 PRELIMINARIES

**Explicit GNNs.** Let $\mathcal{G} = \{\boldsymbol{A}, \boldsymbol{X}\}$ represents an undirected graph, where $\boldsymbol{A} \in \mathbb{R}^{n \times n}$ is the adjacency matrix indicating the relationships between the nodes in $\mathcal{V} = \{\boldsymbol{v}_1, \boldsymbol{v}_2, \ldots, \boldsymbol{v}_n\}$, and $n$ is the number of nodes. The node feature matrix $\boldsymbol{X} = [\boldsymbol{x}_1, \boldsymbol{x}_2, \ldots, \boldsymbol{x}_n] \in \mathbb{R}^{n \times d}$ contains the features of the nodes, with $d$ representing the feature dimension. Conventional (explicit) GNNs [26; 42] feature a learnable aggregation process centered on the message-passing operation within the graph [19]. This process iteratively propagates information from each node to its neighboring nodes. The formal general structure for each layer $l$ is defined as follows:

$$\boldsymbol{H}^{[l+1]} = f_\theta(\boldsymbol{A}, \boldsymbol{H}^{[l]}), \quad \boldsymbol{H}^{[0]} = \boldsymbol{X}, \tag{1}$$

where $\boldsymbol{H}^{[l]}$ represents the hidden node representation, $f_\theta$ denotes the parameters in $l$-th layer. A commonly used GNN is Graph Convolutional Network (GCN) [26], defined as $\boldsymbol{H}^{[l+1]} = \sigma(\hat{\boldsymbol{A}}\boldsymbol{H}^{[l]}\boldsymbol{W}^{[l]})$, where $\boldsymbol{W}^{[l]}$ denotes the weight matrix of the $l$-th layer, $\sigma(\cdot)$ denotes the activation function, $\hat{\boldsymbol{A}} = \tilde{\boldsymbol{D}}^{-1/2}(\boldsymbol{A} + \boldsymbol{I})\tilde{\boldsymbol{D}}^{-1/2}$ represents the symmetric normalized graph matrix. Here, $\tilde{\boldsymbol{D}}$ is a diagonal matrix with $\tilde{\boldsymbol{D}}_{ii} = 1 + \sum_j \boldsymbol{A}_{ij}$. GNNs leverage the above message-passing operation in Eq. 1 to learn useful information. Still, they often involve a limited number of $l$ layers due to over-smoothing [1], making it challenging for GNNs to capture the long-range dependency on graphs.

**Implicit GNNs.** Similar to traditional explicit GNNs, implicit GNNs [20; 33; 7; 12] also involve an aggregation process, with the distinction that the depth of layers (iteration step $k$) is infinite. The aggregation process in IGNNs is typically defined as $\boldsymbol{Z}^{[k+1]} = \sigma(\hat{\boldsymbol{A}}\boldsymbol{Z}^{[k]}\boldsymbol{W} + b_\Omega(\boldsymbol{X})), k = 1, 2, \ldots, \infty$, where $b_\Omega$ represents affine transformation parameterized by $\boldsymbol{\Omega}$, and the weight matrices $\boldsymbol{W}$ and $\boldsymbol{\Omega}$ are globally shared at each iteration step. The IGNN model $f_\theta$ is formally described by

$$\boldsymbol{Z}^\star = f_\theta(\boldsymbol{Z}^\star, \hat{\boldsymbol{A}}, \boldsymbol{X}), \tag{2}$$

where the representation, given as the "internal state" $\boldsymbol{Z}^\star$, is obtained as the fixed-point solution of the equilibrium equation 2. Consequently, the final representation theoretically encompasses information

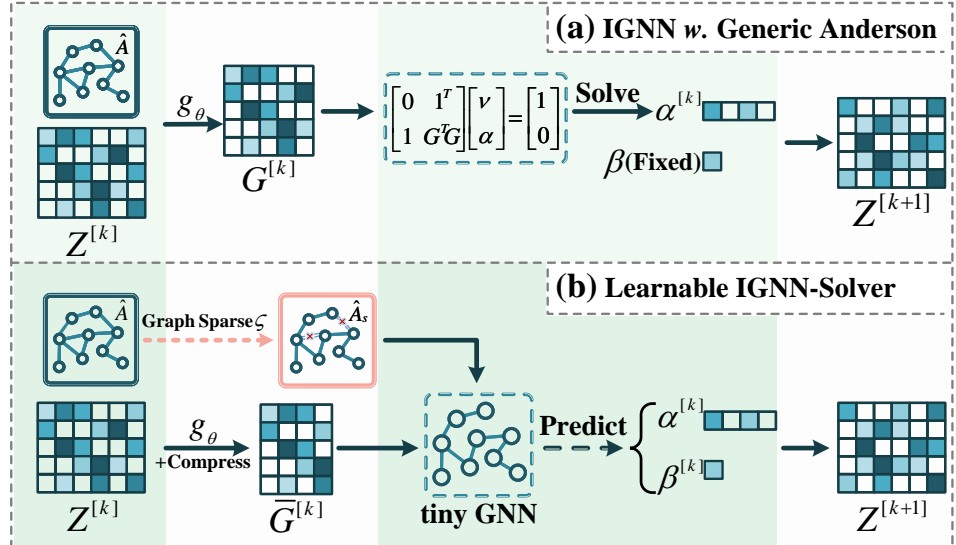

Figure 1: The overall architecture of the IGNN-Solver compared with the generic Anderson solver at each iteration.

from all neighbors in the graph. In practice, IGNNs capture long-range dependencies within the graph, offering better performance compared to GNNs with finite iterations [20; 33]. Another notable advantage of this framework is its memory efficiency, as it only needs to retain the current state $\boldsymbol{Z}$ without requiring additional intermediate representations.

For training parameters $\boldsymbol{\theta} = \{\boldsymbol{W}, \boldsymbol{\Omega}\}$ during the back-propagation process of neural networks, IGNNs compute the Jacobian $\nabla_{\boldsymbol{Z}}\mathcal{L}$ by solving the following equilibrium equation

$$\nabla_{\boldsymbol{Z}}\mathcal{L} = \boldsymbol{D} \odot (\boldsymbol{W}^{\top}\nabla_{\boldsymbol{Z}}\mathcal{L}\hat{\boldsymbol{A}}^{\top} + \nabla_{\boldsymbol{X}}\mathcal{L}), \tag{3}$$

where $\boldsymbol{D} = \phi'(\boldsymbol{W}\boldsymbol{X}\hat{\boldsymbol{A}} + b_{\Omega}(\boldsymbol{U}))$ and $\phi'(\cdot)$ refers to the element-wise derivative of the map $\phi$. Once $\nabla_{\boldsymbol{Z}}\mathcal{L}$ is obtained, we can use the chain rule and the implicit function theorem [28] to easily compute $\nabla_{\boldsymbol{W}}\mathcal{L}$ and $\nabla_{\boldsymbol{\Omega}}\mathcal{L}$.

In summary, IGNNs usually require numerous iterations to obtain equilibrium, resulting in significant overhead during training and inference. Therefore, it is crucial to find a faster and more efficient method for solving the fixed-point equation of IGNNs.

## 4 IMPLICIT GRAPH NEURAL NETWORKS SOLVER

Although traditional fixed-point solvers for IGNNs (as discussed in Section 1 and 2) demonstrate functionality, they are characterized by relatively slow performance, necessitate manual parameter tuning, and are not well-suited for graph-based applications. To address these limitations, we have designed a lightweight, learnable, and content-aware fixed-point solver for IGNN, which integrates a tiny graph neural network, combining the speed advantages of the solver with the information benefits of graph learning. It encodes the relationship between past residuals and the parameters $\alpha^{[k]}$ and $\beta$, allowing for more efficient adjustments of the parameters $\alpha$ and $\beta$ to improve the learning capabilities of the IGNN-Solver, as illustrated in Figure 1.

Due to the characteristics of the implicit model—where its representation capability is independent of the forward computation (i.e., the solver has no knowledge of the original task, such as predicting the node's class, and vice versa)—, we can train this neural solver in a lightweight and unsupervised manner. After the IGNN-Solver is learned, we can utilize it to solve the fixed-point equation of the IGNN layer, thereby further updating the model. The whole training strategy with IGNN-Solver can be found in Appendix B.1 and B.2.

---

**Algorithm 1** IGNN-Solver Iterations and Training

---

**Input:** frozen IGNN model $f_\theta$, initializer $h_\phi$, tiny-GNN predictor $s_\xi$, sparse subgraph $\hat{A}_s$, storage
$\quad G \in \mathbb{R}^{(m+1)\times d'}$.

1: Compute the initial value of fixed point by $Z^{[0]} = h_\phi(X) \in \mathbb{R}^{d'}$
2: Define $g_\theta(Z) = f_\theta(Z) - Z$ and set $G[0] = g_\theta(Z^{[0]})$
3: **for** $k = 0, \ldots, K$ **do**
4: $\quad$ 1) Set $m_k = \min\{m, k\}$ and $G^{[k]} = G\left[0 : (m_k + 1)\right] \in \mathbb{R}^{(m_k+1)\times d'}$
5: $\quad$ 2) Predict $\alpha^{[k]}, \beta^{[k]} = s_\xi(G^{[k]}, \hat{A}_s)$, where $\alpha^{[k]} \in \mathbb{R}^{(m_k+1)}$, $\mathbf{1}^\top \alpha^{[k]} = 1$ and $\beta^{[k]} \in \mathbb{R}^1$
6: $\quad$ 3) Update $Z^{[k+1]} = \hat{\beta}^{[k]} \cdot \mathbf{1}^\top G^{[k]} + \sum_{i=0}^{m_k} \hat{\alpha}_i^{[k]} Z^{[k-m_k+i]}$ $\qquad\qquad \triangleright$ (Anderson step)
7: $\quad$ 4) Update $G = \text{concat}\left(G[1 :], [g_\theta(Z^{[k+1]})]\right)$
8: **end for**
9: **if** it is inference stage **then**
10: $\quad$ **return** $Z^{[k+1]}$
11: **else**
12: $\quad$ **return** $(Z^{[k+1]}, G^{[k]}, \alpha^{[k]}, \beta^{[k]})_{k=0,\ldots,K}$ and $Z^{[0]}$
13: $\quad$ Compute $\mathcal{L}_{\text{total}}$ and back-propagate it to update IGNN-Solver $\{s_\xi, h_\phi\}$
14: **end if**

---

## 4.1 GENERAL FORMULATION

For a given IGNN layer $f_\theta$, input features $X \in \mathbb{R}^{n\times d}$ and graph $\hat{A} \in \mathbb{R}^{n\times n}$, we assume that the equation $Z = f_\theta(Z, \hat{A}, X) = \sigma(\hat{A}ZW + b_\Omega(X))$ has an exact solution at its fixed point. This solution can be obtained using classic solvers (such as Broyden's method) and running multiple iterations to achieve high precision.

The overall structure of the IGNN-Solver is shown in Algorithm 1. Specifically, it learns the parameters $\alpha$ that control the weights among past $m$ steps approximate solution, and $\beta$ that control the weights between past residuals and approximate solution, through a solver $s_\xi$ to accelerate the approximate solution of the next update step. To train the IGNN-Solver, we minimize the joint loss function $\mathcal{L}_{\text{total}}$ by back-propagating through this $K$-step temporal process, which is discussed in Section 4.2. It's worth noting that the original IGNN is frozen (i.e., model parameters $\theta$ are fixed), and only the IGNN-Solver parameters $\xi$ are trained here, so we do not need the ground-truth label $y$ that corresponds to input $x$. This implies that IGNN-Solver can also be fine-tuned during inference after deployment.

### 4.1.1 INITIALIZER.

To accelerate the convergence of predicted values, we propose constructing an initializer $Z^{[0]} = h_\phi(X) : \mathbb{R}^d \to \mathbb{R}^{d'}$ for a rapid and reasonable input-based initial estimate value, rather than simply setting the initial values to $\mathbf{0}$ or random, where $\phi$ are parameters of the initializer. We set the intermediate dimension $d'$ to be significantly smaller than the feature dimension $d$ to reduce the training overhead of the initializer.

### 4.1.2 IMPROVED ANDERSON ITERATIONS WITH TINY-GNN.

In the original Anderson Acceleration (AA) solver, the parameter $\beta$ is predetermined and fixed [2], whereas the parameters $\alpha^{[k]}$s are determined by solving numerous linear equations using the least squares method. Although this method operates adequately, its efficiency and adaptability in optimizing IGNN models are limited. Therefore, we propose introducing a tiny and learnable graph neural network, as

$$\alpha, \beta = s_\xi(G, \hat{A}), \text{ where } s_\xi : (\mathbb{R}^{(m_k+1)\times d'} \times \mathbb{R}^n) \to (\mathbb{R}^{(m_k+1)} \times \mathbb{R}^1), \tag{4}$$

to predict the two parameters instead of setting them as the least squares solution on the past residuals $G$. However, directly using the original graph for modeling would *result in an oversized network* [53; 59], deviating from the initial goal of using a tiny network for prediction. On the other hand, because of *the curse of dimensionality*, directly mapping the data from high to extremely low

dimensions is not appropriate [27]. Therefore, in response to the challenges posed by large-scale graphs, we took into account both Graph Sparsification and Storage Compression in the design of the solver, as detailed below:

**Graph Sparsification.**   Firstly, the number of edges in graph $\hat{A}$ is usually large in practice, as it is affected by the scale of the input (for example, in the ogbn-products dataset, the number of edges in the graph is close to 62M). To maintain the lightweight nature of tiny-GNN and reduce the computational cost of prediction, we propose introducing graph sparsification [24; 36; 55] for a more light graph:

$$\hat{A}_s = \varsigma(\hat{A}, \beta), \tag{5}$$

where $\hat{A}_s$ represents the sparse subgraph of $\hat{A}$, obtained via a graph sparsification operator $\varsigma$, with the parameter $\beta$ controls the sparsity of $\hat{A}_s$. We choose the RPI-Graph approach [53] in this paper, which is a plug-and-play graph sparsification method based on the principle of relevant information. It is worth noting that this subgraph is used only in tiny-GNN predictor $s_\xi$, and not in IGNN model $f_\theta$.

**Storage Compression.**   Besides, considering that $n$ (the number of nodes) is relatively large (for example, $n$ is approximately 169K in the ogbn-arxiv dataset), it is not appropriate to map $s_\xi$ from a high-dimensional space to a very low-dimensional one in equation 4 directly [45; 57; 44] owing to the curse of dimensionality [27]. Therefore, to maintain $s_\xi$ fast and compact, we recommend compress $G_i^{[k]}$ to form a smaller yet still representative version $\bar{G}_i^{[k]}$. Specifically, We map it from $\mathbb{R}^{(m_k+1) \times d'}$ to an appropriate space $\mathbb{R}^{(m_k+1) \times p}$ by multiple layer perception (MLP). Then, the nearest $m_k + 1$ sets of fixed points are merged into one feature matrix:

$$\bar{G}^{[k]} = \mathop{\Big\|}_{i=k-m_k}^{k} \bar{G}_i^{[k]}, \tag{6}$$

where $\|_{i=k-m_k}^{k}$ represents the concatenation operation that stack the compressed storage matrix $\bar{G}_i^{[k]} \in \mathbb{R}^{(m_k+1) \times p}$ in the $k$-th iteration.

Once this smaller but still representative storage matrix $\bar{G}$ obtained by above, we treat it as a $p$-channel matrix and employ a tiny graph neural network layer

$$\alpha^{[k]}, \beta^{[k]} = s_\xi(\bar{G}^{[k]}, \hat{A}_s), \tag{7}$$

to predict the relative weights $\alpha^{[k]}$ assigned to past $m_k + 1$ residual, as well as the AA mixing coefficient $\beta^{[k]}$ at the $k$-th iteration.

In this way, $s_\xi$ shall autonomously learn and adjust these parameters $\alpha^{[k]}$ and $\beta^{[k]}$ based on the previous solver steps and receiving gradients from subsequent iterations. We provide a comparison of different choices for $s_\xi$ in Appendix D, demonstrating that IGNN-Solver improves the convergence path. We also introduce how to train it (see below).

## 4.2   TRAINING THE IGNN-SOLVER

Unlike the neural ODE solver, the fixed-point trajectory of the IGNN is not unique. Therefore, trajectory fitting is not applicable to the IGNN-solver training. Instead, the goal is simply to bring everything as close as possible to $Z^\star$. Formally, given an IGNN-Solver $\{s_\xi, h_\phi\}$ that returns $(Z^{[k+1]}, G^{[k]}, \alpha^{[k]}, \beta^{[k]})_{k=0,...,K}$ and $Z^{[0]}$, we introduce three objectives functions for its training. The complete training algorithm can be referred to in Algorithm 1.

**Initializer Loss Functions.**   To train the initializer, we minimize the distance between the initial guess and the fixed-point by

$$\mathcal{L}_{\text{init}} = \|h_\phi(X) - Z^\star\|_2, \tag{8}$$

for facilitating the subsequent iteration process. Since the initializer directly predicts based on the input $X$ without going through iteration, we separate this loss from other components.

**Reconstruction Loss Functions.** The training of the solver does not require reference to label information or any trajectory fitting. Apart from the loss $\mathcal{L}_{\text{init}}$ necessary for training the initializer above, we introduce reconstruction loss

$$\mathcal{L}_{\text{rec}}^{[k]} = \|\boldsymbol{Z}^{[k]} - \boldsymbol{Z}^\star\|_2, \tag{9}$$

where the reconstruction loss $\mathcal{L}_{\text{rec}}^{[k]}$ aims to make all intermediate predictions $\boldsymbol{Z}^{[k]}$ converge to the accurate fixed point $\boldsymbol{Z}^\star$ as closely as possible.

**Auxiliary Loss Functions.** Although we utilize $\alpha^{[k]}$ and $\beta^{[k]}$ to improve the generic solver, we have empirically found that using an auxiliary loss to guide the solver's prediction of $\alpha$ is beneficial sometimes, especially in the early stages of training. Therefore, in practice, we suggest considering this loss and gradually diminishing it as training progresses (i.e., decay the weight of this loss to 0).

$$\mathcal{L}_\alpha = \sum_{k=0}^{K} \|\boldsymbol{G}^{[k]}\alpha^{[k]}\|_2, \tag{10}$$

The final form of the joint loss function is as follows:

$$\mathcal{L}_{\text{total}} = \lambda_1 \mathcal{L}_{\text{rec}}^{[k]} + \lambda_2 \mathcal{L}_{\text{init}} + \lambda_3 \mathcal{L}_\alpha, \tag{11}$$

where $\lambda_1$, $\lambda_2$ and $\lambda_3$ control the weights of three loss functions mentioned above and elaborate more in Appendix C.3.

## 5 EXPERIMENTS

In this section, we compare the speed/accuracy Pareto curve rather than a single point on the curve of IGNN-Solver with IGNNs and several state-of-the-art (SOTA) GNNs on various graph classification tasks at both node and graph levels. We aim to show that 1) IGNN-Solver can achieve nearly a $1.5\times$ to $8\times$ inference acceleration without sacrificing accuracy, and 2) IGNN-Solver are extremely compact and add little overhead to training.

Specifically, we compare our approach against 12 representative baselines and 2 variants: IGNN *w.* AA (using original AA to accelerate IGNN), and IGNN *w.* NN (using a standard NN solver to replace our proposed graph neural solver for parameter learning) on 9 different-field and real-world node classification tasks, including 4 citation datasets: Citeseer, ACM, CoraFull, ogbn-arxiv, 3 social interaction datasets: BlogCatalog, Flickr, Reddit, and 2 product network datasets Amazon-all, ogbn-products. More details about the datasets are given in Appendix C.1. Notably, to demonstrate the scalability of the proposed model to larger datasets, we use 4 *large-scale datasets*: Amazon-all, Reddit, ogbn-arxiv and ogbn-products, two of which are from the Open Graph Benchmark (OGB) [22][1].

Additionally, in Section 5.2, we demonstrate that the training overhead on IGNN-Solver is minimized relative to the total time overhead on IGNN. In Section 5.3, we provide additional evidence regarding the convergence and generalizability of the IGNN-Solver. In Section 5.4, we conduct an ablation study to validate the stability of the IGNN-Solver and the effectiveness of its components. Experimental setups, descriptions and additional results are detailed in Appendices B, C and D.

### 5.1 PERFORMANCE AND EFFICIENCY COMPARISON

In order to demonstrate the superiority of the IGNN-Solver over IGNNs in terms of both performance and efficiency, we specifically analyze the movement of the speed/accuracy Pareto curve across various datasets rather than concentrating on a single point on the curve, which is depicted in Figure 2 and 6 (see Appendix D.1). All experiments are conducted five times, and the best results are reported. The training procedure details and hyperparameters used in each task are provided in Appendix C.

From Figure 2 (results on large-scale datasets: Amazon-all, Reddit, ogbn-arxiv and ogbn-products), we can observe that: 1) Regarding comprehensive performance and efficiency, IGNN-Solver generally

---

[1]The code to reproduce the results in this section has been uploaded as supplementary material.

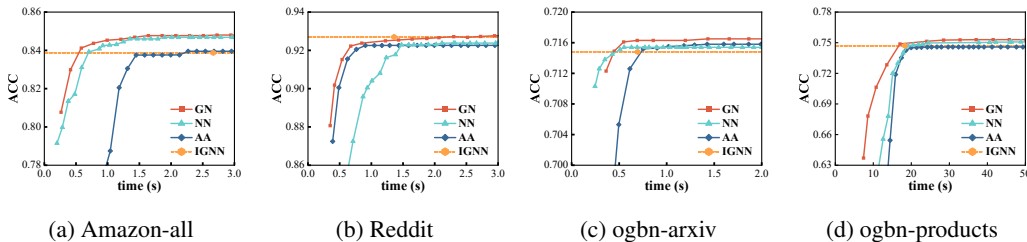

| (a) Amazon-all | (b) Reddit | (c) ogbn-arxiv | (d) ogbn-products |

Figure 2: Comparison of inference time with IGNN-Solvers on 4 large-scale datasets. More results on Citeseer, ACM, CoraFull, BlogCatalog, and Flickr can be found in Figure 6, Appendix D.1. All speed/accuracy curves within the plot are benchmarked on the same GPU with the same experimental setting, averaged over 5 independent runs.

Table 1: Node classification accuracy results on 4 large-scale real-world datasets (more results on small-scale datasets including Citeseer, ACM, CoraFull, BlogCatalog, Flickr can be found in Table 3, Appendix D.1), with experiments conducted over five trials. The mean accuracy ± standard deviation (%) is reported. The **best** and the runner-up results are highlighted in boldface and underline, respectively.

| Type | *Models | Amazon-all | Reddit | ogbn-arxiv | ogbn-products |
|------|---------|------------|--------|------------|---------------|
|      | *Nodes  | $334,863$  | $232,965$ | $169,343$ | $2,449,029$ |
|      | *Edges  | $14,202,057$ | $11,606,919$ | $1,166,243$ | $61,859,140$ |
| Explicit | GCN [26] | $79.12 \pm 1.11$ | $89.65 \pm 0.14$ | $71.56 \pm 0.25$ | $71.91 \pm 0.27$ |
|      | GAT [42] | $76.02 \pm 2.09$ | $90.08 \pm 0.44$ | $71.10 \pm 0.24$ | $72.33 \pm 0.32$ |
|      | SGC [47] | $75.56 \pm 1.75$ | $91.44 \pm 0.41$ | $64.66 \pm 1.02$ | $70.48 \pm 0.19$ |
|      | APPNP [17] | $79.80 \pm 1.22$ | $91.68 \pm 0.24$ | $71.28 \pm 0.29$ | $74.46 \pm 0.64$ |
|      | JKNet [50] | $81.19 \pm 1.09$ | $91.71 \pm 0.31$ | $71.08 \pm 0.35$ | $73.70 \pm 0.58$ |
|      | AM-GCN [45] | $81.85 \pm 1.46$ | $90.20 \pm 0.34$ | $68.83 \pm 0.67$ | $73.19 \pm 0.49$ |
|      | DEMO-Net [48] | $84.08 \pm 1.29$ | $89.53 \pm 0.29$ | $68.40 \pm 0.24$ | $73.88 \pm 0.62$ |
|      | GCNII [11] | $83.60 \pm 2.40$ | $89.87 \pm 0.38$ | $67.66 \pm 0.20$ | $72.88 \pm 0.18$ |
|      | ACM-GCN [35] | $83.04 \pm 2.61$ | $90.37 \pm 0.42$ | $68.32 \pm 0.18$ | $71.68 \pm 0.41$ |
| Implicit | IGNN [20] | $83.90 \pm 0.51$ | $92.30 \pm 1.55$ | $70.49 \pm 0.75$ | $74.63 \pm 0.24$ |
|      | EIGNN [33] | $\underline{84.32} \pm 0.57$ | $92.00 \pm 0.24$ | $70.59 \pm 0.31$ | $74.58 \pm 0.26$ |
|      | MIGNN [7] | $83.68 \pm 0.82$ | $91.98 \pm 0.42$ | $\underline{71.95} \pm 0.44$ | $74.62 \pm 0.32$ |
|      | IGNN *w.* AA | $83.44 \pm 0.19$ | $92.37 \pm 0.35$ | $71.73 \pm 0.41$ | $74.61 \pm 0.28$ |
|      | IGNN *w.* NN | $84.13 \pm 1.04$ | $\underline{92.42} \pm 0.41$ | $70.78 \pm 0.37$ | $\underline{74.69} \pm 0.31$ |
|      | **IGNN-Solver** | $\mathbf{84.50} \pm 0.70$ | $\mathbf{93.91} \pm 0.31$ | $\mathbf{72.53} \pm 0.41$ | $\mathbf{74.90} \pm 0.20$ |

outperforms all other methods, especially in large datasets with a greater graph radius, where this advantage is often more pronounced. This is attributed to the IGNN-Solver's significant improvement in the convergence path and the enhancement of the fixed-point equation's initial point through the initializer; 2) The rapid inference advantage of the IGNN-Solver. For instance, in the large-scale dataset Amazon-all, our IGNN-Solver achieves $8\times$ faster inference speed than IGNN while maintaining the same performance, and there is consistently at least $1.5\times$ acceleration, and a similar pattern has been observed as well on other datasets. Please see more results on other small-scale datasets in Figure 6 of Appendix D.1.

Furthermore, we present the node classification performance results for each method on large-scale datasets(including Amazon-all, Reddit, ogbn-arxiv and ogbn-products) in Table 1 and other small-scale datasets in Table 3 (see Appendix D.1). All experiments are conducted five times, and the average results along with the standard deviation are reported. The training procedure details and hyperparameters used in each task are provided in Appendix C. It can be observed that among all methods, implicit GNNs with infinite depth are often superior to shallow explicit GNNs in most cases. Besides, IGNN-Solver consistently shows higher accuracy percentages across most datasets compared to other SOTA explicit GNNs like DEMO-Net [48], GCNII [11], ACM-GCN [35] etc, which indicates their superior performance.

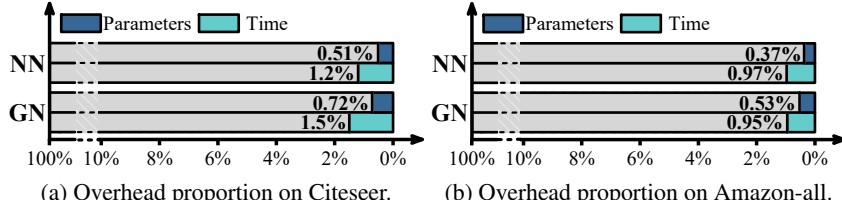

(a) Overhead proportion on Citeseer.  (b) Overhead proportion on Amazon-all.

Figure 3: Relative parameter size and training time of IGNN-Solver on small dataset Citeseer and large one Amazon-all, with similar patterns observed in other datasets.

## 5.2 MODEL SIZE IN TRAINING

To investigate the proportion of training overhead attributed to the solver throughout the entire training process, we depict in Figure 3 the percentage of training computations on IGNN-Solver relative to the total time overhead on IGNN. Our approach is not only effective but also incurs minimal training overhead for the solver: the solver module is relatively small, and requires only about $1\%$ of the original training time required by the IGNN model. Besides, this proportion will be lower for large-scale data. For instance, on the Amazon dataset, IGNN necessitates a total runtime of 3 hours, whereas the solver only requires approximately 1.6 minutes, indicating that our solver maintains its lightweight nature across any-scale datasets.

## 5.3 MORE EFFICIENCY STUDY

We provide additional evidence on the convergence and generalizability of the neural solvers in Figure 4, where we compare the convergence of a pre-trained IGNN model under 1) canonical iteration in IGNN [20]; and 2) IGNN-Solver with $K = 6$.

From Figure 4, we first note that the IGNN-Solver trained with $K$ unrolling steps can generalize beyond $K$, and both solvers eventually reach a stable state, demonstrating good convergence. Secondly, thanks to the fixed-point solution of the graph neural parameterization, we observe that our IGNN-Solver continuously enhances the convergence of the canonical iterators while being more computationally efficient (i.e., each step of the neural solver is cheaper than the standard iterator step). This explains the observed improvement in inference efficiency of at least $1.5\times$ in Section 5.1.

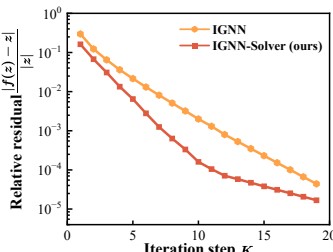

Figure 4: The convergence curve of IGNN-Solver and IGNN on ogbn-arxiv. The former improves the convergence rate.

## 5.4 ABLATION STUDY

Here, we analyze the effect of different loss components of the IGNN-Solver. For the main convergence loss $\mathcal{L}_{conv}$ of fixed-point iterations in equation 11, we designed two contrasting schemes concerning its weight $\lambda_1^{[k]}$s: setting them to a constant value (i.e., $\lambda_1^{[k]} = \lambda_1$ for all $k$), or setting all values except the $K$-th term to zero (i.e., $\lambda_1^{[k]} = \lambda_1$ if $k = K$ else 0). The results of IGNN-Solver and its variants on Citeseer are shown in Figure 5. The patterns observed in other datasets and solvers exhibit similarities.

It is observed that the two variant solvers still perform well, yet our suggested monotonically increasing scheme (i.e., emphasizing the later steps to a greater extent) exhibits the best performance. Additionally, removing the initializer or

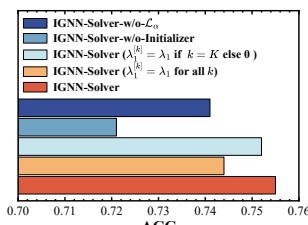

Figure 5: Ablation studies for IGNN-Solver.

the alpha loss $\mathcal{L}_\alpha$ will correspondingly affect performance, with the former having a more detrimental impact on the solver. Nonetheless, the performance of all these ablation settings remains significantly superior to methods without a solver, indicating the advantages of employing a custom learnable solver for the IGNN model.

## 6 CONCLUSION

This paper introduces IGNN-Solver, a novel graph neural solver tailored for achieving fast fixed-point solving in IGNNs. By leveraging the generalized Anderson Acceleration method and parameterizing it with a tiny GNN, IGNN-Solver learns iterative updates as a graph-dependent temporal process. Our extensive experiments demonstrate the significant acceleration in inference achieved by IGNN-Solvers, with a speedup ranging from $1.5\times$ to $8\times$, all while maintaining accuracy. Notably, this acceleration is particularly pronounced as the scale of the graph increases. These findings underscore the potential of IGNN-Solver for large-scale end-to-end deployment in real-world applications.

**Limitations.** In contrast to previous works on IGNNs [20; 33; 7], the theoretical guarantee of fixed-point existence and stability in IGNN-Solver remains uncertain, notwithstanding favorable empirical findings in Figure D.2. This is because we make no specific constraint on the formulation of IGNNs, a departure from the approach adopted in previous IGNN-related studies.

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

# Supplementary Material

## IGNN-Solver: A Graph Neural Solver for Implicit Graph Neural Networks

## A   ANDERSON ACCELERATION SOLVER FOR DEEP EQUILIBRIUMS (DEQS)

---

**Algorithm 2** Anderson Acceleration Solver

---

**Input:** fixed-point function $f_\theta : \mathbb{R}^n \to \mathbb{R}^n$, max storage size $m$, residual control parameter $\beta$

1: Set initial point value $\boldsymbol{Z}^{[0]} \in \mathbb{R}^n$                   ▷ Set $\boldsymbol{0}$ or random value normally

2: **for** $k = 0, \ldots, K - 1$ **do**

3:      1) Set $m_k = \min\{m, k\}$                 ▷ Storage of the most recent steps

4:      2) Solve $\alpha^{[k]} = \arg\min_{\alpha \in \mathbb{R}^{m_k+1}} \left\| \boldsymbol{G}^{[k]}\alpha \right\|_2$, s.t. $\boldsymbol{1}^\top \alpha^{[k]} = 1$      ▷ Compute weights

5:      3) $\boldsymbol{Z}^{[k+1]} = \beta \sum_{i=0}^{m_k} \alpha_i^{[k]} f_\theta(\boldsymbol{Z}^{[k-m_k+i]}) + (1-\beta) \sum_{i=0}^{m_k} \alpha_i^{[k]} \boldsymbol{Z}^{[k-m_k+i]}$      ▷ AA update

6: **end for**

---

Given an implicit network layer $f_\theta$ and an input $\boldsymbol{X}$, DEQs aim to find the fixed point $\boldsymbol{Z}^*$ in the system that represents the equilibrium state of the network output by solving a root-finding problem:

$$g_\theta\left(\boldsymbol{Z}^\star, \boldsymbol{X}\right) := f_\theta\left(\boldsymbol{Z}^\star, \boldsymbol{X}\right) - \boldsymbol{Z}^\star = 0. \tag{12}$$

In practical solving of DEQs, it is often necessary to utilize fixed-point solvers during the training process, such as the classical Broyden method [9] or Anderson Acceleration (AA) [2], which can directly seek the equilibrium point $\boldsymbol{Z}^\star$ through quasi-Newton methods. This approach demonstrates super-linear convergence properties.

We briefly introduce AA-solver here, as our approach relies on it. Algorithm 2 is a pseudocode example illustrating it, where $\boldsymbol{G}^{[k]} = [g_\theta(\mathbf{z}^{[k-m_k]}) \ldots g_\theta(\mathbf{z}^{[k]})]$ is the past residuals. It can be observed that the basic idea of Anderson Acceleration (AA) is to accelerate the generation of the next approximate solution by utilizing the information stored from the history $m$ iteration steps, constructing a normalized linear combination with weights $\alpha^{[k]}$. It computes the weights greedily to minimize the linear combination of past AA update steps. This approach is typically particularly effective for iteratively slow-converging processes, such as solving implicit functions.

Subsequently, during the backward process, the implicit function theorem [28] can be employed to implicitly differentiate through the equilibrium point and generate gradients with respect to the model parameters $\theta$ by solving linear equations based on the Jacobian matrix. Since the computation process relies only on the final output and does not require storing any intermediate activation values [4], the memory consumption during the training process remains constant, which is also a key reason why DEQs are attractive.

## B   TRAINING STRATEGIES FOR IGNN USING IGNN-SOLVER.

### B.1   IGNN TRAINING WITH FROZEN IGNN-SOLVER

During the training processing of the IGNN model, we assume that both the initializer $h_\phi$ and the neural solver $s_\xi$ of IGNN-Solver have approached the desired state via the training method outlined in Section 4.1, and they will no longer be updated in the subsequent model training. For a given IGNN layer $f_\theta$, as well as the input graph $\boldsymbol{A}$ and feature matrix $\boldsymbol{X}$, we define:

$$g_\theta(\boldsymbol{Z}^\star, \hat{\boldsymbol{A}}, \boldsymbol{X}) := f_\theta(\boldsymbol{Z}^\star, \hat{\boldsymbol{A}}, \boldsymbol{X}) - \boldsymbol{Z}^\star = 0, \tag{13}$$

and utilize the IGNN-Solver proposed in Section 4.1 to solve equation 13, obtaining the fixed point $\boldsymbol{Z}^\star$. Furthermore, the fixed point $\boldsymbol{Z}^\star$ is passed through a linear layer to obtain an embedding $\boldsymbol{H} = \text{Linear}(\boldsymbol{Z}^\star)$. Then we jointly with downstream tasks, compute the Binary Cross Entropy (BCE) loss [15] to back-propagate and update the parameters of $f_\theta$ in a supervised way. The complete process is illustrated in Algorithm 3.

---

**Algorithm 3** IGNN model training

---

**Input:** graph matrix $\hat{A}$, feature matrix $X$, fixed IGNN-Solver $s_\xi$ and initializer $h_\phi$

1: Initialize $Z^{[0]} = h_\phi(X)$ and randomly initialize $f_\theta$
2: **while** Stopping condition is not met **do**
3:      $Z^\star \leftarrow$ Solve $f_\theta(Z^\star, \hat{A}, X) - Z^\star = 0$        $\triangleright$ via frozen $s_\xi$, $\hat{A}$ and initial value $Z^{[0]}$
4:      $H \leftarrow$ Linear($Z^\star$)
5:      $\mathcal{L}_{\text{task}} \leftarrow$ BCE($H, Y$)        $\triangleright$ computing the label loss
6:      Back-propagate $\mathcal{L}_{\text{task}}$ to update $f_\theta$
7: **end while**
8: **return** $f_\theta$

---

## B.2 ADVANCED IGNN TRAINING STRATEGY WITH IGNN-SOLVER

Given the fact that model parameter $\theta$ gradually updates only with the model training iterations (i.e., we assume $\|\theta_{t+1} - \theta_t\|$ is only related to $\theta_t$ at training step $t$), we propose training the lightweight IGNN-Solver $s_\xi, h_\phi$ and the large model $f_\theta$ in an alternating way. Specifically, we adopt the following procedure:

(i) Warm up and train the IGNN model and its solver (IGNN-Solver) for some steps.

(ii) Fix the IGNN-Solver $s_\xi, h_\phi$ and solving the fixed points of $f_\theta$ in IGNN via Algorithm 3 for $T_1$ steps.

(iii) Fix the current model parameters $\theta$ and start fine-tuning the IGNN-Solver $s_\xi, h_\phi$ via Algorithm 1 over some $T_2$ steps.

(iv) Repeat steps (ii) and (iii) until reaching the maximum training steps for the IGNN model.

Here $T_1$ is approximately 4% to 10% of $T_2$, adjusted as needed. The sum of all $T_2$ values is referred to as epoch$_{\text{max}}$. It is reassuring that the additional cost of fine-tuning the IGNN-Solver (step (iii)) is sufficiently offset by the substantial benefits of accelerated solving (step (ii)). Moreover, we are surprised to find that the proposed IGNN-Solver does not exhibit a decline in expressive capability for excessively small or large $T_1$ values. This indicates that, thanks to its robust stability, accidentally setting $T_1$ high does not significantly affect the normal training of the IGNN. Conversely, if $T_1$ is set slightly lower, the model still demonstrates good generalization capabilities. Thus, the IGNN-Solver shows low sensitivity to $T_1$.

## C EXPERIMENTAL DETAILS

The experiments are conducted on 9 public datasets of different scales, including 5 widely adopted benchmark datasets Citeseer, ACM, CoraFull, BlogCatalog, Flickr, and 4 large-scale benchmark datasets Amazon-all, Reddit, ogbn-arxiv and ogbn-products. We endeavor to conduct inference testing under almost identical hyperparameter conditions as previous work [7; 33; 20; 45; 26; 42; 47; 17; 50], including performance and efficiency comparisons. All the Experiments are conducted independently five times (i.e., using 5 random seeds) on a machine with Intel(R) Xeon(R) Gold 6138 CPU @ 2.00GHz with a single 3090 GPU. A detailed description of all tasks and additional details are provided below.

### C.1 DATASETS

- **Citeseer** [26]: This dataset is a citation network of research papers, which are divided into six categories. The citation network takes 3,327 scientific papers as nodes and 4,732 citation links as edges. The feature of each node in the dataset is a word vector to describe whether the paper has the corresponding words or not.

- **ACM** [54]: It is a citation network dataset, where nodes represent papers and node features are constructed by the keywords. Papers are divided into 3 categories according to their types of conferences.

Table 2: Hyper-paramaters setting on training. *In addition, we decay the loss weight $\lambda_1$ from 0 to the setting value on a linear schedule over all IGNN-Solver training steps.

| Scale | Datasets | nhid | dropout | lr | $K$ | epoch$_{\max}$ | training | test | $\lambda_1$* | $\lambda_2$ | $\lambda_3$ |
|---|---|---|---|---|---|---|---|---|---|---|---|
| Small | Citeseer | 128 | 0.5 | 0.002 | 10 | 100 | 360 | 1000 | 0.1 | 5 | 1e-4 |
| | ACM | 128 | 0.5 | 0.001 | 10 | 100 | 180 | 1000 | 0.3 | 5 | 1e-4 |
| | CoraFull | 512 | 0.5 | 0.002 | 15 | 300 | 4200 | 1000 | 0.3 | 5 | 1e-4 |
| | BlogCatalog | 512 | 0.5 | 0.001 | 10 | 100 | 360 | 1000 | 0.5 | 5 | 1e-4 |
| | Flickr | 512 | 0.5 | 0.002 | 15 | 200 | 540 | 1000 | 0.1 | 5 | 1e-5 |
| Large | Amazon-all | 128 | 0.5 | 0.005 | 20 | 1000 | 16970 | 28285 | 0.3 | 5 | 1e-5 |
| | Reddit | 512 | 0.5 | 0.005 | 15 | 2000 | 139,779 | 46,593 | 0.5 | 5 | 1e-5 |
| | ogbn-arxiv | 128 | 0.5 | 0.001 | 20 | 2000 | 90,941 | 48,603 | 0.5 | 5 | 1e-5 |
| | ogbn-pruducts | 128 | 0.5 | 0.001 | 20 | 2000 | 196,615 | 2,213,091 | 0.5 | 5 | 1e-5 |

- **CoraFull** [8]: Similar to the Citeseer dataset, CoraFull is a well-known citation network labeled based on the paper topic, which contains 19,793 scientific publications. CoraFull is classified into one of 70 categories, where nodes represent papers and the edges represent citations.

- **BlogCatalog** [37]: This dataset is a social relationship network. The graph is composed of bloggers and their social relationships (such as friends). Node attributes are constructed by keywords in the user profile. The labels represent bloggers' interests. All nodes are divided into six categories.

- **Flickr** [48]: It is a graphic social network where nodes represent users and edges correspond to the friendships among users. All the nodes are divided into 9 classes according to the interest groups of users.

- **Amazon-all** [51]: The widely-used benchmark dataset Amazon-all encompasses the Amazon product co-purchasing network dataset. This dataset represents products as nodes and co-purchases as edges. It includes 58 product types, each with over 5,000 products, selected from a total pool of 75,149 product types.

- **Reddit** [16]: The Reddit dataset is a social network where nodes represent posts, and edges indicate that the same user commented on two connected posts. Each node contains 602 dimensional features.

- **ogbn-arxiv** [22]: The ogbn-arxiv dataset is a citation network between all Computer Science (CS) arXiv papers. Each node is an arXiv paper, and each directed edge indicates that one paper cites another. Each paper comes with a 128-dimensional feature vector obtained by averaging the embeddings of words in its title and abstract. The task is to predict the 40 subject areas of arXiv CS papers.

- **ogbn-products** [22]: The ogbn-products dataset contains an undirected and unweighted graph, representing an Amazon product co-purchasing network. Nodes represent products sold on Amazon, and edges between two products indicate that the products are purchased together. The task is to predict the category of a product in a multi-class classification setup, where the 47 top-level categories are used for target labels.

## C.2 BASELINES

We compare IGNN-Solver with 12 state-of-the-art methods, including 3 implicit IGNNs, i.e., MIGNN [7], EIGNN [33], IGNN [20], and 9 explicit/traditional GNNs, i.e., AM-GCN [45], GCN [26], GAT [42], SGC [47], APPNP [17], JKNet [50], DEMO-Net [48], GCNII [11], ACM-GCN [35]. The specific parameter settings are as follows:

For the configuration of hidden layers, for the sake of fairness, we set the same hidden layers for all baselines. For example, in the Flickr dataset, the dimension of the hidden layers is uniformly set to 512 for all methods (including ours, see in Table 2). In addition, for EIGNN, the arbitrary gamma is set to 0.8, which is consistent with the original paper. For AM-GCN, the number of nearest neighbors in KNN graph is set from $5, 6, 7$. For GCN, the hidden layer setup is same as others. For GAT, three attention headers are used for each layer. For SGC, the power of self-loops in the graph adjacency matrix is set to 3. For APPNP, we set the number of iterations to 3 and teleport

probability $\alpha$ to 0.5. For JKNet, we set layers to 1 in small datasets and set to 8 in large ones. For DEMO-Net, the regularization parameter is set as $5e - 4$, the learning rate is set as $5e - 3$, and the hash dimension is set as 256. For GCNII, we set $\alpha_\ell = 0.1$ and $L_2$ regularization to $5e - 4$, consistent with the original paper. For ACM-GCN, we set layers to 1 in small datasets and set to 4 in large ones. For our IGNN-Solver and its variants, we employ the same settings as the basic IGNN model. Furthermore, we adjust these parameters affecting the convergence and performance of IGNN through a combined approach of hierarchical grid search and manual tuning. The Adam optimizer [25] is used for optimization.

### C.3 HYPER-PARAMETERS SETTING IN IGNN-SOLVER

We present in Table 2 all the hyper-parameters preset by IGNN-Solver across all datasets, where $K$ represents the threshold of maximum iteration in IGNN-Solver, $\text{epoch}_{\max}$ means the maximum training steps in IGNN. It is worth noting that we set the value of $\lambda_1$ in a linearly increasing manner to penalize the intermediate estimation error when the training step $K$ is large. Similarly, we set the value of $\lambda_3$ in a linearly decreasing manner to prevent overfitting $\alpha$ during later training epochs.

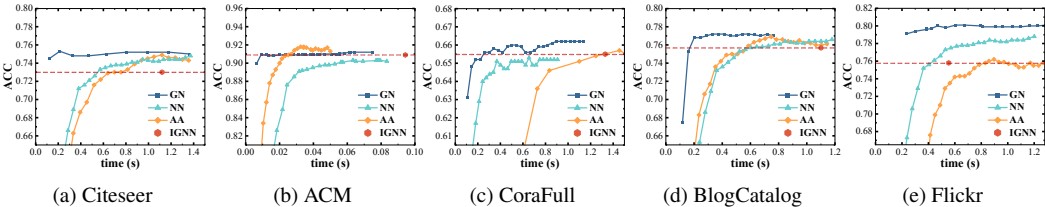

| (a) Citeseer | (b) ACM | (c) CoraFull | (d) BlogCatalog | (e) Flickr |

Figure 6: Comparison of inference time with IGNN-Solvers on 5 small-scale datasets. All speed/accuracy curves within the same plot are benchmarked on the same GPU with the same experimental setting, averaged over 5 independent runs.

Table 3: Node Classification accuracy results on 5 small-scale real-world datasets, with experiments conducted over five trials. The mean accuracy ± standard deviation (%) is reported. The **best** and the runner-up results are highlighted in boldface and underline, respectively.

| Type | *Models | Citeseer | ACM | CoraFull | BlogCatalog | Flickr |
|------|---------|----------|-----|----------|-------------|--------|
|      | *Nodes  | $3,327$  | $3,025$ | $19,793$ | $5,196$ | $7,575$ |
|      | *Edges  | $4,732$  | $13,128$ | $65,311$ | $171,743$ | $239,738$ |
| Explicit | GCN [26] | $64.80 \pm 4.15$ | $83.78 \pm 3.95$ | $35.98 \pm 9.44$ | $71.30 \pm 1.12$ | $76.98 \pm 1.83$ |
|      | GAT [42] | $71.24 \pm 2.88$ | $80.60 \pm 5.12$ | $50.24 \pm 1.87$ | $76.24 \pm 2.76$ | $72.64 \pm 3.23$ |
|      | SGC [47] | $70.32 \pm 2.75$ | $85.40 \pm 1.23$ | $46.56 \pm 4.43$ | $69.76 \pm 1.11$ | $71.88 \pm 3.47$ |
|      | APPNP [17] | $61.56 \pm 8.92$ | $84.16 \pm 4.25$ | $21.24 \pm 4.76$ | $61.34 \pm 9.39$ | $73.42 \pm 3.80$ |
|      | JKNet [50] | $63.78 \pm 8.76$ | $64.96 \pm 5.47$ | $23.04 \pm 6.01$ | $72.62 \pm 6.83$ | $75.68 \pm 1.15$ |
|      | AM-GCN [45] | $73.10 \pm 1.62$ | $89.56 \pm 0.30$ | $53.40 \pm 1.59$ | $73.86 \pm 1.10$ | $76.86 \pm 2.02$ |
|      | DEMO-Net [48] | $68.34 \pm 2.94$ | $84.38 \pm 2.19$ | $61.74 \pm 3.65$ | $74.26 \pm 2.70$ | $75.60 \pm 3.95$ |
|      | GCNII [11] | $71.98 \pm 0.80$ | $85.36 \pm 1.05$ | $57.64 \pm 3.34$ | $74.94 \pm 3.81$ | $79.92 \pm 2.14$ |
|      | ACM-GCN [35] | $72.38 \pm 1.46$ | $88.98 \pm 0.41$ | $59.88 \pm 1.59$ | **$78.18 \pm 1.75$** | $74.82 \pm 3.78$ |
| Implicit | IGNN [20] | $72.96 \pm 1.83$ | $90.88 \pm 0.95$ | $65.52 \pm 0.51$ | $75.68 \pm 0.55$ | $75.80 \pm 0.29$ |
|      | EIGNN [33] | $72.38 \pm 1.36$ | $88.36 \pm 1.03$ | $61.80 \pm 0.60$ | $75.34 \pm 0.38$ | $75.66 \pm 0.94$ |
|      | MIGNN [7] | $73.79 \pm 0.94$ | $89.59 \pm 1.61$ | $62.94 \pm 0.46$ | $76.68 \pm 1.49$ | $74.96 \pm 0.49$ |
|      | IGNN w. AA | $75.28 \pm 0.38$ | **$91.34 \pm 0.46$** | $65.88 \pm 0.34$ | $76.82 \pm 0.34$ | $75.84 \pm 0.27$ |
|      | IGNN w. NN | $74.78 \pm 0.42$ | $91.08 \pm 0.56$ | $65.62 \pm 0.16$ | $76.88 \pm 0.74$ | $78.55 \pm 0.49$ |
|      | **IGNN-Solver** | **$75.60 \pm 0.22$** | $91.20 \pm 1.15$ | **$66.08 \pm 1.23$** | $77.64 \pm 0.54$ | **$80.14 \pm 0.23$** |

## D MORE RESULTS AND ADDITIONAL EXPERIMENTS

### D.1 MORE PERFORMANCE AND EFFICIENCY COMPARISON ON OTHER SMALL-SCALE TASKS

We further test IGNN-Solver for a few small-scale graph node classification tasks, including Citeseer, ACM, CoraFull, BlogCatalog and Flickr. We employ the training procedure details in Appendix C and hyperparameters used outlined in Appendix B, and report the mean accuracy ± standard deviation in Table 3 and Figure 6. Generally, learning LRD is not crucial for these tasks, as the diameter of the

graphs is quite small [7]. However, as seen in Table 3, even for these small-scale node classification tasks, the IGNN-Solver can still surpass IGNN and even outperforms many explicit GNNs and other enhanced implicit ones. As shown in Figure 6, even in small-scale tasks, the solver still improves upon the convergence path of IGNN. These results and which in Section 5.1 confirm the expressive power of the IGNN-Solver using learnable graph neural, even exceeding that of many explicit and improved implicit GNNs.

## D.2 OTHER OPTIONS FOR GRAPH NEURAL LAYERS IN IGNN-SOLVER.

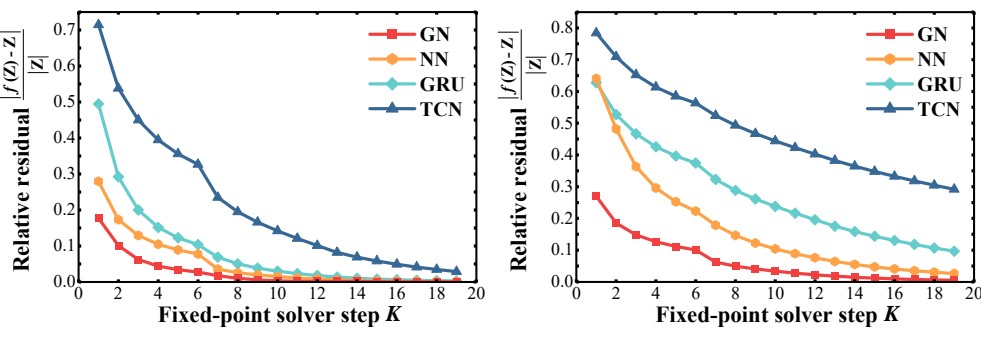

(a) The relative error curve during warm-up.    (b) The relative error curve during inference.

Figure 7: The proposed IGNN-Solver (GN) improves the convergence path, which is attributed to its lightweight nature and capability to leverage rich graph information.

We provide additional evidence on the convergence and generalization of IGNN-Solver, where we tried other options besides our proposed graph neural (GN) layers, including neural network (NN), temporal convolutional network (TCN) [29] and gated recurrent unit (GRU) [14]. We present the relative residual curves on the Citeseer dataset during the warm-up stage and the final inference stage. As illustrated in Figure 7, TCN and GRU, despite having more parameters and more complex structures, actually perform worse than NN. Note that IGNN-Solver (GN) improves the convergence path of all other solvers and exhibits the fastest rate of decrease, proving to be the most suitable solver for IGNN, which is attributed to its capability to leverage rich graph information effectively.

## D.3 TRAINING DYNAMICS OF IGNNS WITH IGNN-SOLVER

In this section, we present the time cost trends of IGNN and IGNN-Solver during training and inference on the Amazon dataset. From Figure 8, it is evident that IGNN starts relatively fast overall at the beginning of training. However, as the training of IGNN progresses, the model becomes increasingly complex, leading to a sharp rise in the time cost for fixed-point computation. After 100 epochs, this cost remains persistently high and fluctuates continuously. This issue arises because the IGNN reaches the maximum number of iterations without achieving the preset minimum error.

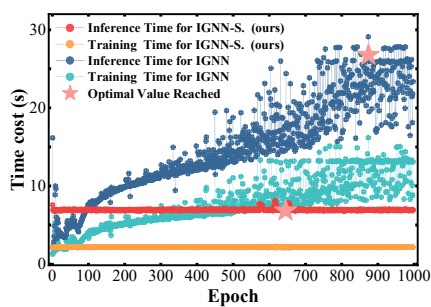

Figure 8: Training time and inference time(s) per epoch of IGNN and IGNN-Solver on Amazon-all datasets.

On the contrary, our IGNN-Solver demonstrates significantly lower training and inference time consumption compared to the former and maintains a stable state throughout the process. This is attributed to the solver's consistent memory consumption and its rapid fixed-point computation capability, which drives the unique advantage of IGNN-Solver.

