# OpenReview forum: "IGNN-Solver: A Graph Neural Solver for Implicit Graph Neural Networks"
_ICLR.cc/2025/Conference — ICLR 2025 Conference Withdrawn Submission_

### Official Review · Reviewer_1QRx · 2024-11-01

**Soundness:** 2
**Presentation:** 3
**Contribution:** 2
**Rating:** 5
**Confidence:** 3

**Summary:**

IGNN can effectively capture long-range dependencies in graphs and mitigate the over-smoothing problem. However, IGNN is limited by the high computational cost of fixed-point iterations. Therefore, this paper proposes the IGNN-Solver algorithm. The algorithm parameterizes the Anderson Acceleration method with a small GNN, effectively accelerating the training and inference process of IGNN. Through experiments on multiple datasets in this paper, the IGNN-Solver algorithm achieves a 1.5x-8x speedup without sacrificing effectiveness. This paper will facilitate the large-scale deployment of IGNN in practical applications.

**Strengths:**

1. Due to the implicit layer of the IGNN, formulated as a fixed-point equation, it can access infinite hops of neighbors implicitly. This enables IGNN to address the long-standing over-smoothing and long-range dependency issues that have plagued explicit GNNs, preventing them from becoming deeper and larger. Therefore, optimizing the computational burden of IGNN, which has stronger scalability, is a potentially promising topic in the journey towards larger graph models. I acknowledge the significance of this paper.

1. This paper integrates the generalized Anderson Acceleration method, graph sparsification, and graph compression algorithms. It employs a multi-dimensional graph acceleration approach, which is concise and clear. At the same time, experiments on multiple graph datasets have been conducted, strongly demonstrating the superiority of IGNN-Solver over IGNN. It alleviates the notorious computational burden of GNNs in both training and inference processes.

1. The proposed IGNN-Solver algorithm effectively accelerates the inference process, achieving a 1.5x-8x speedup without any performance degradation, which is a very impressive result.

**Weaknesses:**

1. Given that the IGNN algorithm can implicitly capture long-range dependencies, I had hoped that the IGNN-Solver would show a greater advantage over explicit GNNs on large datasets. However, on the ogbn-arxiv and ogbn-products datasets, the IGNN-Solver did not demonstrate a significant performance improvement over explicit GNNs. I would like to see not only the comparison of inference times between the IGNN-Solver and traditional IGNN, but also a comparison of the training and inference speeds of the IGNN-Solver with those of traditional explicit GNNs. I hope that the IGNN-Solver can significantly outperform explicit GNNs in terms of performance or efficiency, which would convince me of the practical significance of this work.

1. In Section 4.1.2 of the paper, "IMPROVED ANDERSON ITERATIONS WITH TINY-GNN" is mentioned. I would like to know more specifically the impact of having this module on the acceleration of inference.
Additionally, we all know that for most graph datasets, there is a significant performance drop after reducing 50% of the edges. I would like to know the specific settings of the GNN Sparsification part in IGNN-Solver and the impact of this part's pruning rate on the overall effect.

1. The results in Figure 2 and Table 1 seem to have some inconsistencies. Is this due to different experimental settings? For example, in Figure 2(c), the best accuracy of the IGNN-Solver is around 0.716, while the accuracy given in Table 1 is 0.725. This is not a big issue; I just hope the authors can pay attention to such minor flaws.

1. The accuracy of the baselines in the ogbn-products dataset seems a bit low. According to the official LeaderBoards of OGB and my experimental experience, the accuracy of algorithms such as GCN/GAT/GraphSAGE on ogbn-products is generally between 0.76-0.8. These are all obviously higher than the corresponding values given in Table 1 and the accuracy of the IGNN-Solver.

In summary, if the authors can address my concerns, I would be open to increase my score.

**Questions:**

As mentioned in the "Weaknesses" section.

---

### Official Review · Reviewer_4wCP · 2024-11-03

**Soundness:** 2
**Presentation:** 2
**Contribution:** 2
**Rating:** 3
**Confidence:** 4

**Summary:**

This paper introduces a novel graph neural solver designed to enhance the efficiency of implicit graph neural networks (IGNNs). The proposed approach aims to facilitate rapid fixed-point computation. The authors provide empirical evidence demonstrating that their solver achieves significant acceleration, yielding several-fold speedups without compromising predictive performance on several benchmark graph datasets.

**Strengths:**

1. The paper is well-written, clear, and easy to comprehend.

2. The experimental results show the proposed neural solver can achieve significant acceleration for IGNN, yielding several-fold speedups without compromising predictive performance on several benchmark graph datasets.

**Weaknesses:**

1. **Lack of Theoretical Justifications:** The paper does not provide sufficient theoretical underpinnings for the proposed fixed-point neural solver. Notably, it remains unclear if the fixed-point equation is well-posed, and there are no convergence guarantees for the solver.

2. **Uncertain Effectiveness of the Learnable Initializer:** Without guarantees of convergence, it is ambiguous whether the learnable initializer can indeed reduce the number of iterations required, thus raising questions about its potential to speed up the optimization process.

3. **Limited Architectural Verification:** The experimental validation of the proposed neural solver is restricted to the IGNN architecture. It is uncertain whether its benefits extend to other implicit GNN frameworks.

4. **Absence of Ablation Studies:** The paper does not include ablation studies that assess the solver’s ability to capture long-range dependencies, a key advantage associated with implicit GNNs.

5. **Over-Smoothing Mitigation:** The study does not address whether the proposed solver can mitigate the over-smoothing issue often encountered in GNNs.

**Questions:**

On Lines 269-270 and 288-289, the authors argue that mapping data from a high-dimensional space to an extremely low-dimensional space is inappropriate due to the curse of dimensionality. Could you provide a more detailed explanation of this claim and its implications?

---

### Official Review · Reviewer_veBM · 2024-11-03

**Soundness:** 2
**Presentation:** 2
**Contribution:** 2
**Rating:** 5
**Confidence:** 3

**Summary:**

This paper introduces a novel approach to improving the efficiency of implicit graph neural networks.  The authors proposes IGNN-Solver.  It uses a learnable initializer to estimate the initial point, and then use a tiny GNN working on a sparsified graph to predict the coefficient used in the generalized Anderson Acceleration updating step.

**Strengths:**

IGNN-Solver introduces a low overhead in the training procedure (1% - 2% of the total training time).

**Weaknesses:**

1. The expreriment results in Table 1 cannot convince me.  At least, the baselines for ogbn-arxiv and ogbn-products is severely sandbagged.  For example, on ogb learderboard, GCNII is 72.74 for ogbn-arxiv, and GCN is 75.64 on ogbn-products.  These gaps are more than 4% compared with the number reported in the paper.

2. There is no study on how accurate the alpha predicted by the tiny GNN model.

**Questions:**

1. What is the architecture of the initializer?

2. Figure 2 is confusing.  What does these legends mean?

---

### Official Review · Reviewer_qjDY · 2024-11-04

**Soundness:** 2
**Presentation:** 2
**Contribution:** 2
**Rating:** 3
**Confidence:** 4

**Summary:**

The paper introduces IGNN-Solver, a novel graph neural solver for implicit graph neural networks (IGNNs). IGNNs have strong expressive power with a single layer, but suffer from slow inference speeds due to the computationally expensive fixed-point iterations required to solve the equilibrium equation. IGNN-Solver addresses this limitation by leveraging a tiny graph neural network to predict the next fixed-point iteration, significantly accelerating inference without sacrificing accuracy.

**Strengths:**

1.	The paper demonstrates that IGNN-Solver can achieve a significant speedup in inference time compared to regular IGNNs, and the additional training cost of the IGNN-Solver is minimal.

**Weaknesses:**

1. Current background lacks coverage of recent advances, such as:
Method [1], which models IGNN as a bilevel optimization problem, achieving significant speedups.
Method [2], a scalable implicit model with higher accuracy on the ogbn-arxiv dataset. Given [2] is already cited, consider a comparison to highlight IGNN-Solver's distinct advantages in context.

2. Since efficiency is a core advantage of IGNN-Solver, it is crucial to benchmark its runtime against multiple existing methods beyond the basic IGNN. This will provide a clearer view of IGNN-Solver’s efficiency benefits.

3. To convincingly demonstrate superiority, include a straightforward baseline of IGNN using phantom gradients, a common efficiency enhancement. This comparison would clarify IGNN-Solver's performance against well-known alternatives.

4. Typo and Logic:

Line 093: "[34] ... Subsequently, [43] introduces ...". The chronology is incorrect, as [34] was published after [43].
Equation Formatting: In Equation 4, R^n -> R^{n\times n}

[1] Zhong Y, Vu H, Yang T, et al. Efficient and Effective Implicit Dynamic Graph Neural Network[C]//Proceedings of the 30th ACM SIGKDD Conference on Knowledge Discovery and Data Mining. 2024: 4595-4606.
[2] Liu J, Hooi B, Kawaguchi K, et al. Scalable and Effective Implicit Graph Neural Networks on Large Graphs[C]//The Twelfth International Conference on Learning Representations. 2024.

**Questions:**

Please refer to the weaknesses

---

### Official Review · Reviewer_jA1E · 2024-11-10

**Soundness:** 3
**Presentation:** 3
**Contribution:** 2
**Rating:** 6
**Confidence:** 4

**Summary:**

This paper presents IGNN-Solver, a novel approach to accelerate fixed-point solving in implicit graph neural networks (IGNNs), addressing the scalability challenges posed by traditional IGNNs. Using a generalized Anderson Acceleration method parameterized by a GNN, IGNN-Solver models iterative updates as a graph-dependent temporal process. Experiments show that IGNN-Solver achieves a 1.5× to 8× speedup in inference with no loss in accuracy, enabling efficient performance on large-scale graphs.

**Strengths:**

1. This paper tries to answer a very important question: how to accelerate Implicit Graph Neural Networks (IGNNs), which is of interest in the community. IGNNs have some advantages over traditional GNNs, while IGNNs suffer from slow training and inference speed. And this hinders the usage of IGNNs in many applications, espcially when graphs are large.
2. The high-level idea of the proposed solver is cleary demonstrated in Figure 2, which make it easy to understand.
3. Although the novelty of the method is not that strong, the proposed model shows the good empirical results on node calssification task and outperform the vanilla IGNNs.

**Weaknesses:**

1. More deeper analysis on why the proposed solver can be faster than others. In my view, the speedup comes from the less number of iterations required. I think that would be better if the authors can provided some theoretical analysis. If theoreical anlaysis is difficult to have, I would like to see some empirical evidences on how many iterations the proposed solver needs vs the traditional solver needs.
2. The high-level descriptions on the method (RPI-Graph) used for graph sparsification are not provided. I think it would be better if the authors can explain some high-level idea on this sparsification method. It can make the article more self-contained.
3. In my view, this work is mainly about a plug-in solver for Implicit GNNs, not specifically for that model named IGNNs. Therefore, I think that it would be better to apply the proposed solver to different implicit GNNs, such as MIGNN.

**Questions:**

1. I am a bit confused about how Figure 2 the inference speed/accuracy Pareto curve is drawed? Given a dataset and a mode/solver, the inference time should roughly remain the same for different runs. Why the acc can increase when we spend more time on the inference? Is that about adjusting the number of itertions used in the solver to see the different inference times?
2. On about the training dynamics of IGNNs with IGNN-Solver, I would like to ask why the inference/training time increases as the number of epochs increase. I think IGNNs have a maximum number of iterations. If IGNNs always reach the maximum number of iterations, in the end, the train/inference of IGNN should be constant (like a flat line in the last few epochs). Any explanations on this?

---

### Note · Authors · 2024-11-25

I have read and agree with the venue's withdrawal policy on behalf of myself and my co-authors.